# Synthesis of Oligomeric Silicone Surfactant and its Interfacial Properties

**Da Yin [1], Pingya Luo [1], Jie Zhang [2],\* , Xuyang Yao [3], Ren Wang [2], Lihui Wang [2] and Shuangwei Wang [2]**

1   State Key Laboratory of Oil and Gas Reservoir Geology and Exploitation, Southwest Petroleum University, Chengdu 610500, China; yida-tlm@petrochina.com.cn (D.Y.); luopy@swpu.edu.cn (P.L.)
2   Drilling Fluids Research Department, CNPC Engineering Technology R&D Company Limited, Beijing 102206, China; wangrdr@cnpc.com.cn (R.W.); wanglhdri@cnpc.com.cn (L.W.); wangswdr@cnpc.com.cn (S.W.)
3   CNPC Xinjiang Oilfield Company, Karamay 834000, China; yaoxydr@cnpc.com.cn
\*   Correspondence: zhangjiedri@cnpc.com.cn; Tel.: +86-10-8016-2085

**Abstract:** During the exploitation of low permeability gas-condensate reservoirs, the mud filtrate, acidizing liquid, and fracturing fluid invade the reservoir and condensate gas, severely reducing the permeability of the reservoirs due to increased capillary pressure and water wettability. For the current paper, an oligomeric silicone surfactant (OSSF) containing sulfonic acid groups was synthesized to improve the flowback of such fluids. The critical micelle mass concentration and critical surface tension were determined by equilibrium surface tension. The surface tension increased with the hot rolling temperature and decreased with the addition of NaCl, KCl, or $CaCl_2$. When the concentration exceeded critical micelle mass concentration, a micelle was formed and its size increased with mass concentration. OSSF adsorption through solid–liquid surface changed the surface chemical composition of the cores and transferred the wettability of cores from water-wet to preferential gas-wet by decreasing the surface energy. At the same time, the increasing temperature led to a change in the adsorption isotherm of quartz sand from Langmuir type (L-type) to "double plateau" type (LS-type) in the OSSF solution. In addition, NaCl decreased the relative foam volume of OSSF while extending the half-life. OSSF decreased the initial foaming volume and stability of the induction period and accelerated sodium dodecyl benzene sulfonate (SDBS) formation.

**Keywords:** silicone surfactant; oligomeric; surface tension; wettability; adsorption; foam performances

## 1. Introduction

Many low permeability gas-condensate reservoirs possess small reservoir pores and throats wetted by water, which can generate high capillary pressure. The mud filtrate, acidizing liquid, and fracturing fluid invasion during the drilling process cannot entirely flow back, so the reservoir water saturation increases, as the gas relative permeability significantly decreases near the wellbore. The reduction of surface tension of invaded fluids is the most effective way to decrease the damage caused by water blocking. As the temperature and pressure drop below the dew point near the wellbore during drilling, a special gas accumulates and condensates into liquid. Since gas condensate cannot spread on the pore surface wetted by water, it results in the throat being plugged leading to a sharp decrease in gas relative permeability. The rate of such permeability damage to the reservoir can reach 70–90%, with the reduction of the gas well production of less than one-third [1–3]. Numerous methodologies, including solvent injection, microwave heating, and hydraulic fracturing, have been

used to reduce condensed liquid damage [4–6]. Furthermore, wettability alteration from water-wet to preferential gas-wet is also employed to boost gas condensate flow [7].

Capillary pressure is the pressure between two immiscible fluids flowing through a narrow space, resulting in the interaction of forces between the fluids and the solid walls of the tube. Moreover, a decrease in capillary pressure can be achieved by using an adsorbing surfactant at the gas–liquid interface [8–10].

Gas reservoir wettability is the ability of the mud filtrate, formation water, and gas condensate to stick to and spread onto the pores, which influence various fluid flow and irreducible water saturation by decreasing capillary pressure and surface energy. Wettability alteration can be achieved by using an adsorbing surfactant at the solid–liquid interface [11–13]. Since the solid–liquid and gas–liquid interfaces appear simultaneously in the flow back of invaded mud filtrate, capillary pressure reduction and preferential gas-wetness alteration can be achieved with the addition of a surfactant to the drilling fluid.

Surfactants are organic compounds composed of hydrophilic polar and hydrophobic non-polar groups, with only a small amount necessary to significantly change the interfacial properties. Currently, hydrocarbon and fluorocarbon surfactants are the most commonly utilized surfactants in oil and gas development. Hydrocarbon surfactants, such as Emulsifier OP-10, Sodium dodecyl benzene sulfonate (or ABS in short), and Lauryl benzene sulfonate diethanolamine salt (or ABSN in short), are inexpensive and environmentally friendly, they can reduce the surface tension between gas and liquid to 30–35 mN/m, and they are widely used in emulsions, foaming agents, and capillary pressure reduction. However, they are unable to alter the wettability of reservoirs from water-wet to preferential gas-wet by adsorption. On the other hand, fluorocarbon surfactants are expensive and hazardous to the environment but possess good surfactivity, which can decrease the surface tension between gas and liquid below 20 mN/m, promoting defoaming and wettability alteration.

Silicone surfactants have excellent thermal stability, high surface activity, and outstanding ecological safety and biodegradability, which are mainly attributed to the feature of the siloxane chain [14–16], and they have a very high molecular weight [17]. Reports have shown that sulfonate silicon surfactants can be synthetized via free radical copolymerization of polysiloxanes and sulfonic acid monomers. The hydrophobicity and oleophobicity of the surfactant increase with the polysiloxane chain length. The hydrophobic groups are cross-linked with relatively low silicon content. Due to siloxane hydrolysis, the interfacial tension of the sulfonate trisiloxane surfactant, prepared by hydrosilylation reaction, increases after a prolonged period of storage at normal temperature [18]. However, the modification of polyether trisiloxanes via sulfonation of the epoxy group exhibits excellent chemical stability, such as acid resistance, alkali resistance, and salt resistance, but also exhibits a short siloxane chain, as well as being expensive and employing toxic chloroplatinic acid as a catalyst in the hydrosilation reaction [19]. The phosphate fluorosilicone surfactant synthesized by hydrosilylation and esterification of hydrogen-containing fluorine polysiloxane, polyethylene glycol monoallyl ether, and phosphoric acid displays excellent low surface tension, critical micelle concentration, and a good resistance to acid, alkali, and salt. When hydrosilicone oil is the raw material, the molecular weight of synthetic surfactant is very high, but it also uses the expensive and toxic chloroplatinic acid catalyst [20].

Siloxane surfactants have been rarely applied to the development of oil and gas reservoirs, despite their wide and extensive industrial usage. Current studies on silicone surfactants mainly focus on low and high molecular copolymers, such as trisiloxane and modified silicone oil surfactants. In this work, a newly designed oligomeric surfactant bearing a siloxane spacer and sulfonic group was synthesized. The surfactant properties necessary in low permeability gas-condensate reservoirs development were investigated, including surface tension, wetting ability, and foaming property. To the best of our knowledge, our study is only one of a rare few that examined the interfacial properties of oligomeric silicone surfactant for the development of gas reservoirs. Our accumulated results provide a theoretical basis for expanding the application of silicone surfactants in low-permeability condensate reservoirs.

## 2. Experimental Design

### 2.1. Materials

Diethoxydimethylsilane (DDS) and vinyltriethoxysilane (VTES) were purchased from Aladdin Industrial Co. Ltd. (Shanghai, China), chemical pure. 2-Acrylamide-2-methyl propane sulfonic acid (AMPS) was purchased from Beijing Chemical Co. Ltd. (Beijng, China), chemical pure. Azobisisobutyronitrile (AIBN), alcohol, sodium hydroxide, hydrochloric acid, and ethylene glycol were purchased from Xilong Chemical Co. Ltd. (Guangzhou, Guangdong, China), analytical grade. Triply distilled water was used in all experiments. The cores used in this study were Yingxi limestone from Qinghai Oilfield (Huatugou, Qinghai, China) and Haian sandstone from Haian Petroleum Instrument Co. Ltd. (Haian, Jiangsu, China).

### 2.2. Synthesis of Vinylorganosilicon Oligomeric (VOGO)

The synthesis procedure is as follows: diethoxydimethylsilane (80.00 g, 0.539 mol) and ethanol (40.00 g, 0.868 mol) were placed into a four-necked flask equipped with a stirrer, reflux condenser, and thermometer. The mixture was stirred at room temperature for 15 min and then heated to 60 °C. Hydrochloric acid (2.80 M, $H_2O$) was then slowly added to the reaction mixture via syringe. After 1 h of stirring at 60 °C, vinyltriethoxysilane (4.00 g, 0.021 mol) was added dropwise to the reaction mixture over 30 min and stirred for an additional 1 h. A green transparent mixture was obtained when cooled to room temperature. This process is shown in Scheme 1, Steps 1 and 2.

Fourier-transform infrared spectroscopy (FT-IR) (KBr pellet): 3126.16 $cm^{-1}$ (-Si-CH₃, stretching bending vibration), 1043.55 $cm^{-1}$ (Si-O-Si, stretching vibration), 1595 $cm^{-1}$ ($CH_2$=CH-Si, stretching vibration).

Gel permeation chromatography (GPC): $M_n = 2.3 \times 10^3$, $M_p = 1.6 \times 10^3$, $M_w = 2.5 \times 10^3$, $M_z = 3.0 \times 10^3$, $M_w/M_n = 1.12$, $M_z/M_n = 1.32$.

### 2.3. Synthesis of Oligomeric Silicone Surfactant (OSSF)

The synthesis procedure is as follows: azobisisobutyronitrile (0.08 g, 0.487 mmol) was added to ethanol (20.00 g, 0.499 mol) and stirred until fully dissolved (solution 1). 2-Acrylamide-2-methyl propane sulfonic acid (3.00 g, 0.014 mol) was added to a NaOH solution (10.00 mol/L, $H_2O$) and stirred until fully dissolved. Ethanol (20.00 g, 0.499 mol) was then added (solution 2). The VOGO was placed into a four-necked flask equipped with a stirrer, reflux condenser, and thermometer, then solutions 1 and 2 were added dropwise into the flask at 60 °C for 30 min, respectively. Then, the pH of the mixture was adjusted using a NaOH solution (10.00 mol/L, $H_2O$) to approximately 5–6 and stirred at 65 °C for 4 h. The solvent was then evaporated using rotary evaporation. The precipitate was filtered, giving the target compound. This process is shown in Scheme 1, Step 3. The product was analyzed using FT-IR and GPC.

FT-IR (KBr pellet): 3419.35 $cm^{-1}$ (-N-H, stretching vibration), 1654.84 $cm^{-1}$ (C=O, stretching vibration), 1072.58 $cm^{-1}$ (-S=O, asymmetry stretching vibration), 1191.86 $cm^{-1}$ (-S=O, symmetry stretching vibration).

STEP 1:

STEP 2:

STEP 3:

**Scheme 1.** Synthesis of the oligomeric silicone surfactant (or OSSF in short).

*2.4. Characterization Methods*

The methods for measurement and determination of the properties of OSSF are described in this section.

FT-IR. FT-IR was recorded using an FT-IR spectrometer (Thermo Fisher Scientific, Waltham, MA, USA). Measurements were performed by dispersing samples in anhydrous KBr pellets.

Determination of molecular weight of VOGO. The average molecular weight of the oligomeric was determined by GPC (Waters, model Xevo TQ, Milford, MA, USA) with a refractive index detector using PL gel column (Agilent Technologies Inc., Palo Alto, CA, USA) in tetrahydrofuran with a flow rate of 1 mL/min. The tested mass concentration of OSSF in the THF solution was 0.5%, and its intrinsic viscosity was 1.12 dL/g at 30 °C. The molecular weights were calculated with a calibration relative to polystyrene standards.

Measurement of surface tension. Surface tension measurements were performed using an interfacial tensiometer Sigma 701 (KSV, Helsinki, Finland). A dynamic Wilhelmy plate method was used to perform each series of measurements. The mass concentration was manually changed with microsyringes. Each value was the average of the three measurements. In addition, critical micelle mass concentration was determined by the cross point of the two lines before and after critical micelle mass concentration on the $\gamma$-$\rho_n$ curve.

Measurement of thermal stability. OSSF solutions with a pH value adjusted to approximately 9–10 using a NaOH solution (10.00 M, $H_2O$) were first heat-rolled for 16 h in a roller oven (Haian Petroleum Scientific Research Instrument, Jianngsu, China), and then the surface tension of the heat-rolled solutions was measured at 25 °C.

Measurement of salt tolerance. OSSF was added to the NaCl, KCl, and $CaCl_2$ solutions, respectively. The solution was first stirred well and then allowed to stand for 24 h at room temperature indoors. Subsequently, the surface tension of the solution was measured at 25 °C.

Measurement of the size and distribution of micelles. The size and distribution of micelles in the OSSF solutions were measured using a laser nanometer analyzer (Malvern Instruments, Malvern, UK) at 25 °C.

Core preparation. The cores were prepared in the following steps: first, the cores were cut into thick slices (about half a centimeter). OSSF was dissolved in water (OSSF solution). Then, the cores slices were cleaned with water and evacuated for several hours in a vacuum chamber. Next, the slices were steeped in 100 mL of the OSSF solution, whose pH value was adjusted to approximately 9 to 10 using a NaOH solution (10.00 M, $H_2O$). The mixture was then heat-rolled for 8 h at 150 °C in a roller oven. Finally, the cores slices were dried in a vacuum drier at 150 °C for 4 h and then cooled to 25 °C.

Contact angle measurement and surface energy determination. Contact angle was formed between the plane tangent to the surface of the cores and liquids (water and ethylene glycol) at the wetting perimeter. It was measured by a JC200D3 contact angle meter (Zhongchen Digital Technology Equipment, Shanghai, China) and calculated using the five-point fitting method. Following the Owens-Wendt geometric theory, the surface energy ($\gamma_s$) of the cores was calculated by Equations (1) and (2) [21,22]:

$$(1 + \cos\theta)\gamma_L = 2\left(\gamma_s^d \gamma_L^d\right)^{1/2} + 2\left(\gamma_s^p \gamma_L^p\right)^{1/2} \tag{1}$$

$$\gamma_s = \gamma_s^d + \gamma_s^p \tag{2}$$

where $\theta$ (°) is the contact angle between the core surface and liquid; $\gamma_L$ (N/m) is the free energy of the liquid surface; $\gamma_S$ (N/m) is the free energy of the core surface; $\gamma_s^d$ (N/m) is the core dispersion force; $\gamma_L^d$ (N/m) is the liquid dispersion force; $\gamma_s^p$ (N/m) is the core polar force; and $\gamma_L^p$ (N/m) is the liquid polar force.

Measurement of adsorption isotherms. OSSF was dissolved in water generating the OSSF solution. Clean 90–100 mesh size quartz sand was first added to the OSSF solution in order to maintain a 1:20 solid–liquid ratio. The pH value of the solution was adjusted to approximately 9–10 using a 10 M NaOH aqueous solution and then heat-rolled for 8 h in a roller oven. After filtrating the quartz sands, the mass concentration of OSSF was measured using an ultra-violet and visible spectrophotometer (INESA Instrument, Shanghai, China). The amount of adsorption ($\Gamma$) on the surface of the quartz sands was calculated using Equation (3) [23]:

$$\Gamma = \frac{\rho_s - \rho_0 - \rho_b}{m} \tag{3}$$

where $\rho_s$ (mg/L) is the mass concentration of OSSF in the solution before adsorption; $\rho_0$ (mg/L) is the mass concentration of OSSF in the blank solution after adsorption; $\rho_b$ (mg/L) is the mass concentration of OSSF in the solution after adsorption; $V$ (L) is the volume of the OSSF solution; and $m$ (g) is the mass of the quartz sands.

Measurement of foaming property. Two solutions were prepared: OSSF was dissolved in water and a 5.00 wt % NaCl solution was added to the OSSF solution. Then, 100 mL of each solution was separately stirred at 1200 r/min for 3 min on a high-speed mixer with variable frequency (Tongchun Petroleum Instruments, Qingdao, China) to form a foaming solution. Both foam solutions were poured into two 1000 mL measuring cylinders. The relative foam volume (RFV) and foam half-life (FHL) in the measuring cylinders with time were recorded.

Measurement of defoaming property. Sodium dodecyl benzene sulfonate (SDBS) was dissolved in water generating a SDBS solution with 5 g/L concentration. Then, 100 mL of this solution was stirred at 1200 r/min for 3 min on a high-speed mixer with variable frequency creating the foaming solution. The foam solution was poured into a 1000 mL measuring cylinder. After 1 mL of the OSSF solution was added dropwise onto the foam in the measuring cylinder, the change of the relative foam volume in the measuring cylinder with time was recorded.

Measurement of foam-suppressing property. To 100 mL of the above SDBS solution, OSSF was added. The mixture was stirred at 1200 r/min for 3 min on a high-speed mixer with variable frequency, and the foam solution was then poured into a 1000 mL measuring cylinder. The change in foam volume in the measuring cylinder with time was recorded.

## 3. Results and Discussion

### 3.1. Surface Tension

Surface tensions of OSSF aqueous solutions were measured to evaluate their surface activities. Figure 1 shows the surface tension of OSSF ($\gamma$) versus mass concentration ($\rho_n$) of the aqueous solutions of OSSF at 25 °C. It can be seen that the surface tension of the OSSF solution decreases initially with increasing mass concentration, which indicates that all surfactants are adsorbed at the air–water interface. The appearance of a plateau suggests that micelles are formed. The mass concentration corresponding to the intersection point (0.980 g/L) is regarded as critical micelle mass concentration. The surface tension corresponding to the intersection point (20.631 mN/m) is regarded as critical surface tension.

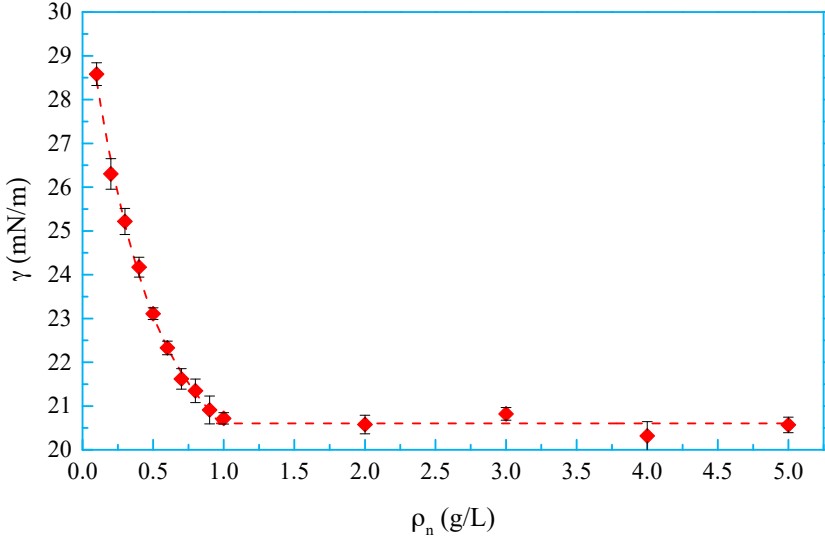

**Figure 1.** Surface tension of OSSF versus mass concentration (T = 25 °C, free pH).

### 3.2. Thermal Stability

It is necessary for the surfactant to have good chemical stability and thermal tolerance in the development of gas reservoirs. In order to promote the dispersion of bentonite clay and organic drilling fluid additive, preventing corrosion of the drill pipe and casing pipe, and to inhibit the dissolution of calcium and magnesium salts, the drilling fluid is usually weakly alkaline, with pH between 8 and 11. However, OSSF silanol (Si-OH) groups can interreact at high temperature in an alkalis environment. Hot rolling testing is generally adopted to study the influences of high temperature and shear on the stability of the drilling fluid component, whose flow rate in the roller heating furnace can reach 0.16 m/s.

The surface tension of the OSSF solution versus mass concentration under various hot rolling temperatures is presented in Figure 2. A decrease in the surface tension of the surfactant aqueous

solution is observed and tends to stabilize at the hot rolling temperature when the mass concentration rises from 1.00 to 5.00 g/L. However, under the same mass concentration, higher hot rolling temperature leads to a greater surface tension in the OSSF solution. Notably, based on accelerative hot rolling tests, the surface tension of the OSSF solution is still lower than 25 mN/m below 150 °C.

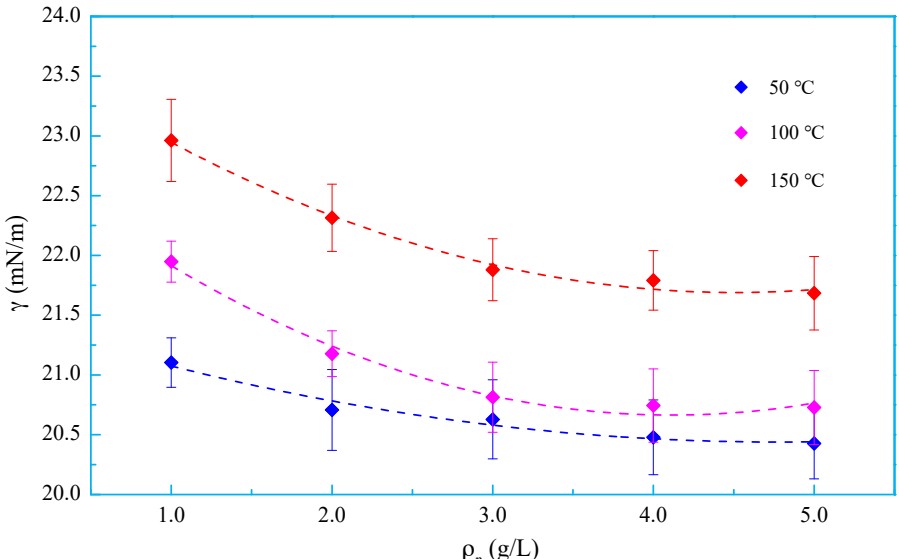

**Figure 2.** Surface tension of OSSF versus mass concentration under various hot rolling temperature (T = 25 °C, pH = 9–10).

### 3.3. Salt Tolerance

The surfactant must exhibit high salt tolerance in high-salinity reservoirs. Figures 3–6 display the relationship between surface tension ($\gamma$) and weight percentage (wt %) of OSSF in NaCl, KCl, and CaCl$_2$ solutions at 25 °C. Experimental results show that the surface tensions of the same weight percentage of OSSF solutions decrease with increasing weight percentages of NaCl, KCl, and CaCl$_2$ (Figures 3–5). However, under the same weight percentage, the OSSF solutions' surface tensions decrease with increasing OSSF weight percentage. Figure 6 shows the effect of inorganic salt on the surface tension of the OSSF solution, where the descending order is 10.00 wt % KCl, 10.00 wt % NaCl, and 0.50 wt % CaCl$_2$. Since the inorganic ions can adsorb around the surfactant molecules by electrostatic interaction, the charge number and concentration of inorganic ions that coexist in solution highly affect the hydration layer generated by the charged, polar, and non-polar groups. The p-d$\pi$ coordination bond of the -SO$_3{}^{2-}$ groups can enhance *S* ability to attract electrons from *OH*, promoting an increase in the hydratability of OSSF while decreasing its ability to attract cations. The compressed diffused double layer ultimately weakens the repulsion between the surface molecules, creating a more closely arranged grouping on the interface, which reduces the surface tension of the OSSF solution from a macroscopic view [24,25].

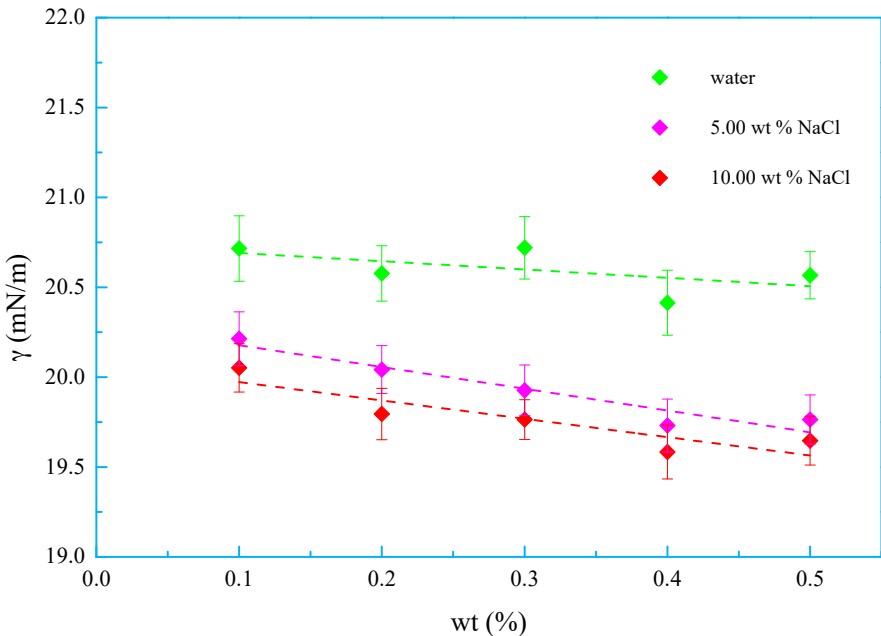

**Figure 3.** Surface tensions of OSSF versus weight percentage in a NaCl solution (T = 25 °C, free pH).

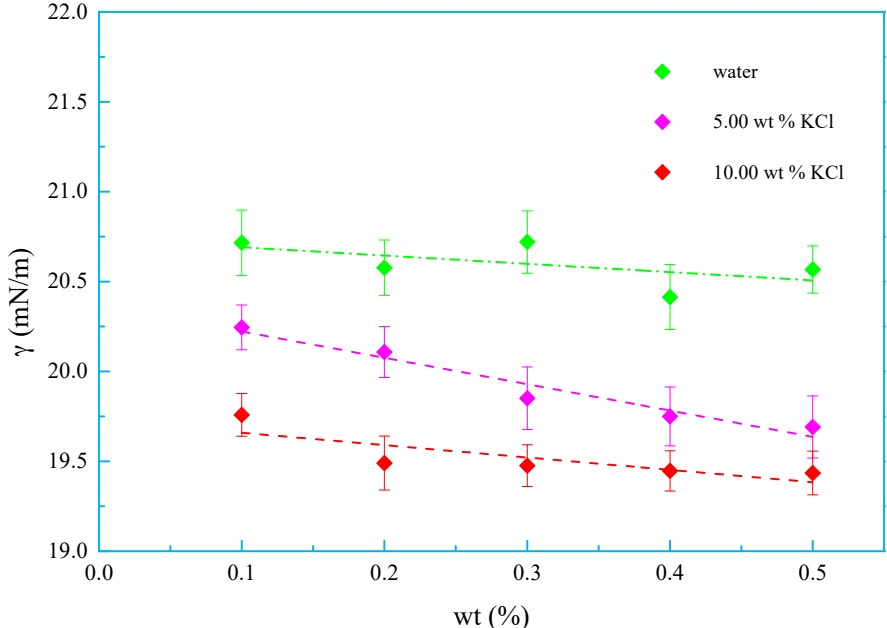

**Figure 4.** Surface tensions of OSSF versus weight percentage in a KCl solution (T = 25 °C, free pH).

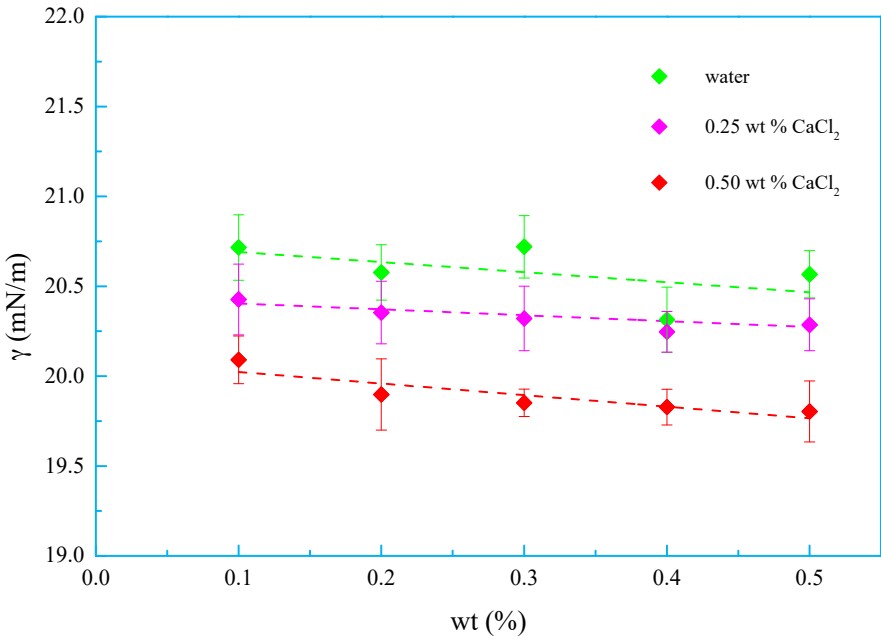

**Figure 5.** Surface tensions of OSSF versus weight percentage in a CaCl$_2$ solution (T = 25 °C, free pH).

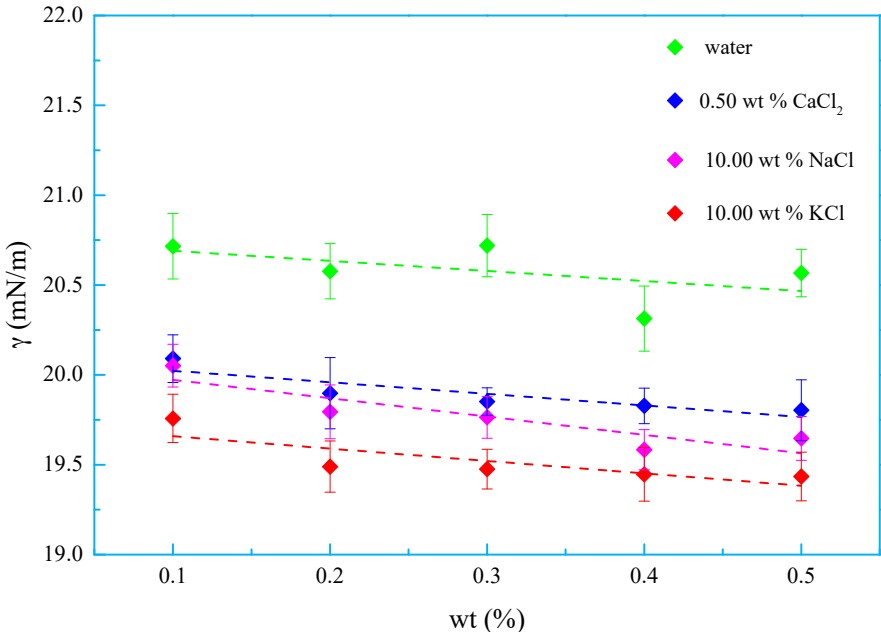

**Figure 6.** Surface tensions of OSSF versus weight percentage in an inorganic salt solution (T = 25 °C, free pH).

### *3.4. Size and Distribution of Micelles*

The mass concentration of the surfactants in an aqueous solution is higher than the critical micelle mass concentration when the surface of the solution is saturated with surfactant molecules. The remaining surfactant molecules in solution form micelles by association under attractive interactions between hydrophobic groups. Figure 7 shows the size and distribution of OSSF micelles versus mass concentration at 25 °C.

According to Figure 7, the siloxane chains of OSSF aggregates form multi-core aggregations in the aqueous solutions when the mass concentration is higher than the critical micelle mass concentration. The size and distribution of OSSF micelles increase with OSSF mass concentration, due to small-sized micelles colliding together and generating larger aggregated micelles by self-assembly. This further

confirms that, when the weight percentage of OSSF is greater than 0.10 wt %, the adsorption of OSSF is saturated on the gas–liquid interface at 25 °C.

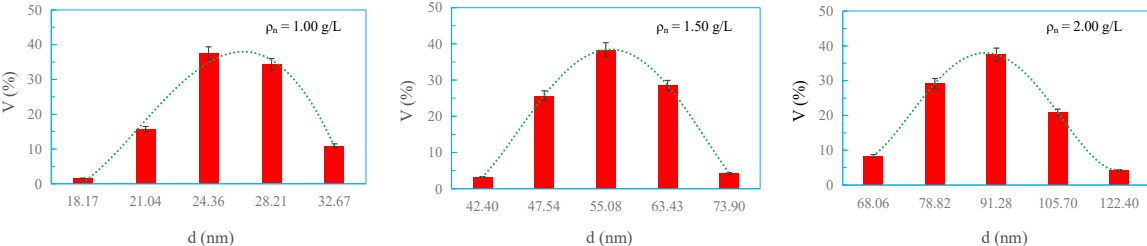

**Figure 7.** Size and distribution of OSSF micelles versus mass concentration (T = 25 °C, free pH).

### 3.5. Contact Angle and Surface Energy

Reservoir grains are hydrophilic in character, but their surface wettability can change from hydrophilic to hydrophobic due to the adsorption of certain surfactants. When the contact angle of water on the rocks is <90°, the reservoir hydrophilicity is generally better than the lipophilicity, known as water wettability. However, when the contact angle of water is greater than 90° but smaller than 180°, the reservoir has a stronger lipophilic capability, meaning that the water can only generate adhesional wetting on the reservoir surface rather that immersional wetting, which can only turn a partial gas–solid interface into a solid–liquid surface, known as preferential gas wettability relative to the water. In terms of the condensate gas reservoir, if the reservoir surface wettability changes from hydrophilic to hydrophobic, gas condensate can spread over the surface occupied by gas, allowing it to flow into the wellbore by spontaneous imbibition. Natural cores are the cylindrical rock samples taken from the reservoir by coring, which are the most important, first-hand, and frequently used materials in oil exploration and geology research. Due to the difficulty in obtaining natural cores and their complex components, artificial cores are used in the laboratory experimental evaluation of reservoir research. As the concentration of the surfactant at the solid–liquid interface is higher than the aqueous solution, the wettability can change by the absorption of surfactant in the solution. Table 1 represents the contact angles on the cores and the surface energy of cores versus weight percentage of OSSF at 25 °C.

**Table 1.** Contact angles on the cores and surface energy of cores versus weight percentage of OSSF (room temperature, free pH).

| Cores | Weight Percentage/wt % | Contact Angles/° | | Dispersion Force/(mJ/m$^2$) | Polar Force (mJ/m$^2$) | Surface Energy/(mJ/m$^2$) |
| --- | --- | --- | --- | --- | --- | --- |
| | | Water | Ethylene Glycol | | | |
| Haian | 0.0 | infiltration | infiltration | - | - | - |
| Yingxi | 0.0 | 13.47 | spreading | - | - | - |
| Haian | 0.1 | 46.86 | 11.32 | 16.53 | 35.09 | 51.63 |
| Haian | 0.2 | 109.35 | 25.28 | 2.96 | 20.55 | 23.51 |
| Haian | 0.3 | 108.76 | 24.82 | 2.30 | 19.80 | 22.09 |
| Haian | 0.4 | 110.12 | 27.14 | 3.47 | 20.80 | 24.27 |
| Haian | 0.5 | 110.01 | 25.37 | 3.91 | 21.58 | 25.49 |
| Yingxi | 0.1 | 52.21 | 12.67 | 21.48 | 26.93 | 48.42 |
| Yingxi | 0.2 | 114.34 | 39.58 | 5.87 | 20.98 | 26.85 |
| Yingxi | 0.3 | 113.61 | 38.72 | 4.97 | 20.37 | 25.34 |
| Yingxi | 0.4 | 113.94 | 39.20 | 5.33 | 20.59 | 25.92 |
| Yingxi | 0.5 | 114.53 | 39.19 | 6.44 | 21.52 | 27.96 |

Table 1 describes the weight percentage of OSSF aqueous solutions, showing that when it is higher than 0.20 wt %, the contact angles of water and the surface energy on the surface of the cores adsorbing OSSF are greater than 90° and less than 30 mJ/m$^2$, respectively. Therefore, OSSF can

transfer the cores' surface property from water-wet to preferential gas-wet by decreasing its surface energy. The contact angles of water significantly increase when the mass fraction changes from 0.10 to 0.20 wt %. Accounting for surface inhomogeneities of these cores, when the weight percentage of the OSSF solutions is greater than 0.20 wt %, the contact angles and the surface energy show insignificant change, which means that a higher OSSF weight percentage cannot improve the preferential gas wettability. These results show that mass fraction plays a major rule in wettability alteration and affects the interfacial viscosity and microscopic distribution of oil, gas, and water in the intergranular pores, which in turn alter the fluid flow properties.

Figures 8 and 9 show that the contact angles of water on the cores which adsorbed OSSF slowly decline, stabilizing within one minute, which indicates that OSSF absorbed on the cores implied wettability alteration of the cores maintained for a certain period. In the case of Yingxi cores steeped in a 0.50 wt % OSSF solution, the contact angles of water on the cores slowly decline from 114.53 to 109.34° in a minute time period, whereas Haian cores decline from 110.07 to 106.46°. However, this testing method has its own defect—compared to the amount of OSSF adsorbed on the cores surface before drying, it could overestimate the adsorbed amount after evaporating the water.

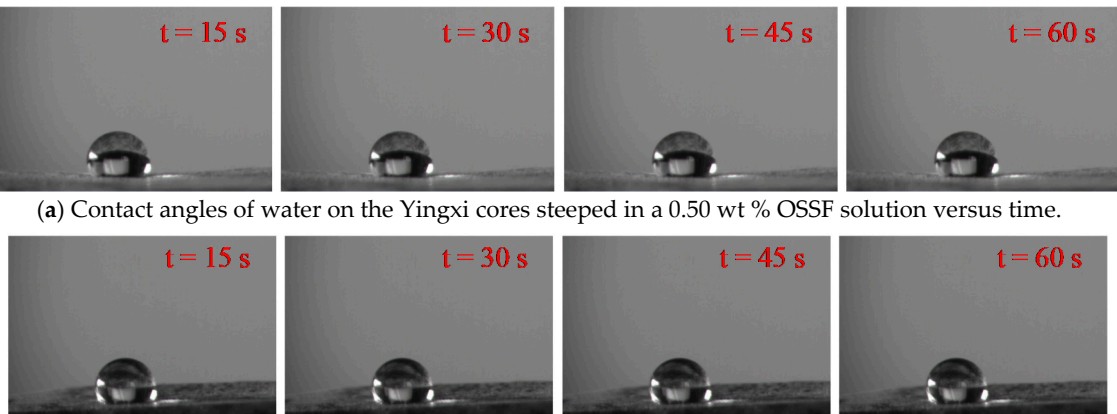

(**a**) Contact angles of water on the Yingxi cores steeped in a 0.50 wt % OSSF solution versus time.

(**b**) Contact angles of water on the Haian cores steeped in a 0.50 wt % OSSF solution versus time.

**Figure 8.** Contact angles of water on the cores steeped in a 0.50 wt % OSSF solution versus time (room temperature, free pH).

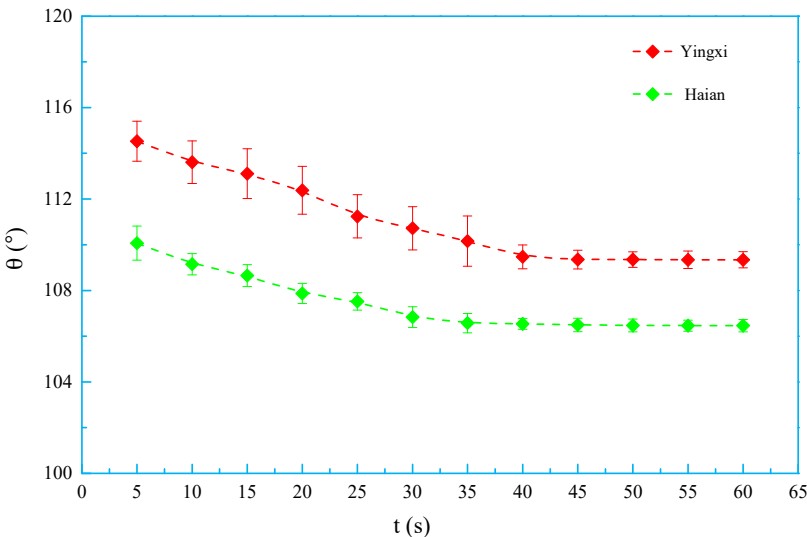

**Figure 9.** Contact angles of water on the cores steeped in a 0.50 wt % OSSF solution versus time (room temperature, free pH).

Wetting is the ability of a liquid to maintain contact with a solid surface, which stems from intermolecular interactions. The energy of a solid relates to the reaction between atoms, molecules, and ions, including covalent bonds, ionic bonds, metallic bonds, van der Waals forces, and hydrogen bonds. In this case, the cores are mainly constructed using chemical bonds. The surface energy of the cores is so high that water can achieve complete wetting. However, the molecules that OSSF adsorbed on the core surface employ physical forces, so the surface energy reduces, allowing only partial wetting by water.

### 3.6. Surface Element Content

Surfactants can change the surface chemical composition by chemical or physical absorption. The adsorption capacity and speed largely depend on the surfactants' active group, but layer thickness is mainly determined by the length of non-attractive segments. Figure 10 represents the elements on the surface of the cores before and after OSSF adsorption.

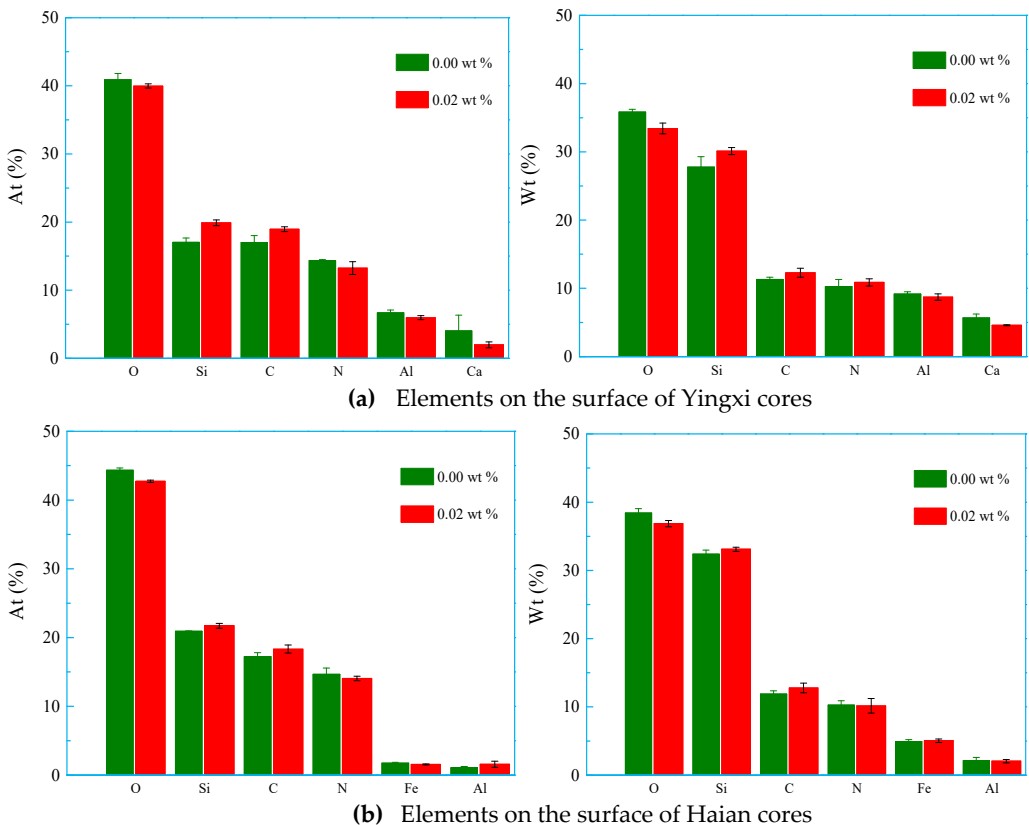

**Figure 10.** Elements on the surface of cores before and after adsorbing OSSF.

Figure 10 shows that for both kinds of cores, the atomic weight and number of Si and C on the core surface absorbing OSSF are significantly higher than cores bereft of OSSF absorbing. Furthermore, OSSF can decrease the atomic weight and number of O on the cores' surface, due to OSSF hydrophilic group absorbing on the cores' surface, generating the oriented silicon-containing (methyl) layer. Therefore, the atomic weight of the surface chemical element is greater than the number of atoms.

### 3.7. Adsorption Isotherms

Since the atoms on the surface of rock are in the residual force field, it can adsorb surfactant molecules in solution. The ultraviolet absorption spectrum curve of the OSSF solution has an absorbance peak at 273 nm and a sub-peak at 375 nm (Figure 11). The absorption coefficient of the OSSF solution at 273 nm is 3.18 L·cm$^{-1}$·g$^{-1}$. Figure 12 shows the standard plot equation $y = 0.0031x$

$-$ 0.0013 ($y$ represents the absorbance at 273 nm and $x$ represents the mass concentration of OSSF); the correlation coefficient square between OSSF mass concentration and measured absorbance is 0.9948. The good linear dependence reveals that this standard plot can be utilized to measure OSSF mass concentration in solution. The adsorbed amount of quartz sands increases to tend a stable value with the increase of mass concentration of OSSF at the same temperature (Figure 13). However, when the mass concentration of OSSF is constant, it increases with the increasing temperature of the solution. The adsorption isotherms of the quartz sands show Langmuir type (L-type) in the OSSF solution at 25 °C. The results indicate that the quartz sands adsorb OSSF primarily via physical adsorption, which is consistent with Henry's law. OSSF in an aqueous solution can adsorb on the surface of quartz sands by hydrogen bonding and dispersion forces until almost all adsorption sites are occupied. However, the adsorption isotherms gradually change to "double plateau" type (LS-type) with increasing solution temperature. This is because at high temperature in an alkalis solution, Si–O–Si groups on the surface of silicate minerals hydrolyze to Si–OH groups, which react with OSSF Si–OH groups at the solid–liquid interface by condensation [26]. It confirms the existence of both physical and chemical adsorption during these processes, with chemical adsorption dominating. Notably, the original physical adsorption amount can accelerate the subsequent chemical adsorption, which increases adsorption after the first platform. Therefore, it is reasonable to consider that the adsorption layer increases with the solution temperature at pH = 9–10.

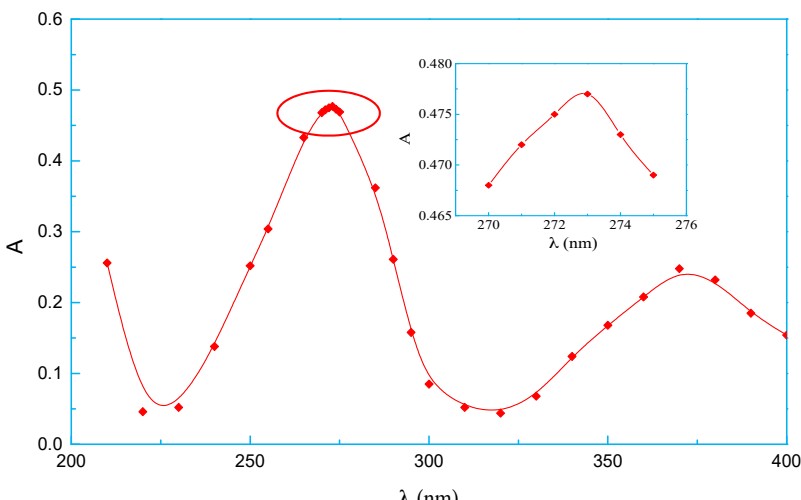

**Figure 11.** Ultraviolet absorption spectrum curve of the OSSF solution.

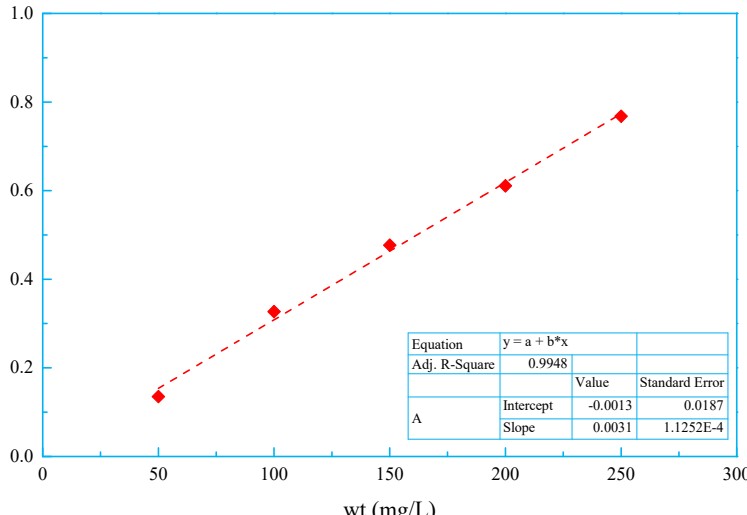

**Figure 12.** Standard plot of mass concentration of OSSF versus its absorbance value at 273 nm.

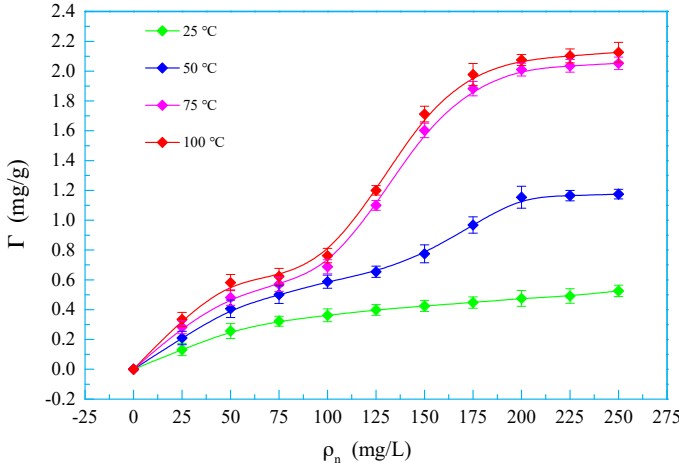

**Figure 13.** Adsorption isotherms of quartz sands in the OSSF solution at different weight percentages (T = 25, 50, 75, and 100 °C, pH = 9–10).

### *3.8. Foaming and Defoaming Property*

### 3.8.1. Foaming Property

The foaming property of the surfactant is closely associated with the dosage of defoamer as well as fluid column pressure, fluid rheology, and pump efficiency [27]. Figure 14 shows that the relative foam volume of OSSF increases to tend a stable value with increasing weight percentage of OSSF in water and 5.00 wt % NaCl solution. On the other hand, Figure 15 shows that the half-life of OSSF first decreases sharply and then plateaus with the increasing weight percentage of both solutions. Despite a similar foaming property, both adhere to this rule at different values. The relative foam volume of OSSF in water is greater than in the 5.00 wt % NaCl solution, but the opposite affect is observed in the foam half-life period. Moreover, the foaming capacity and foam half-lives of OSSF in water and 5.00 wt % NaCl solutions are below 22% and 2 min at the weight percentage <0.5%, respectively. The hydrophobic and hydrophilic groups play a major role in the foaming property. Furthermore, due to the strong hydration ability of the sulfonic acid groups, it can form the stable aqueous layer in the bubble film, but the permethylated siloxane can prevent the diffusion of water molecules and reduce water content in the bubble film, so the forming foam is small and vulnerable to breakdown [28].

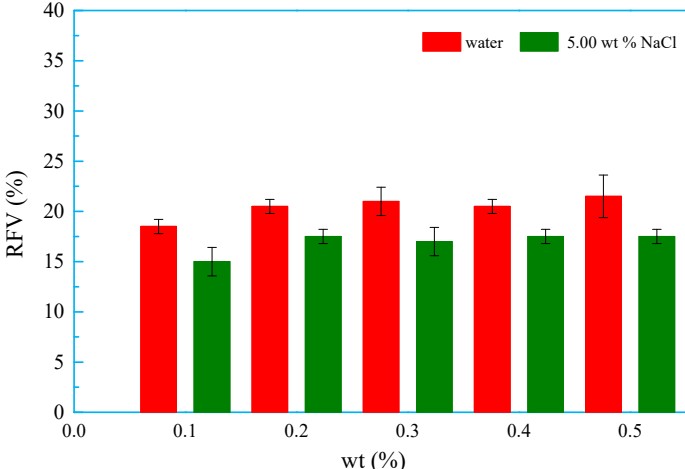

**Figure 14.** Relative foam volume of OSSF versus weight percentage in water and NaCl solution (T = 25 °C, free pH).

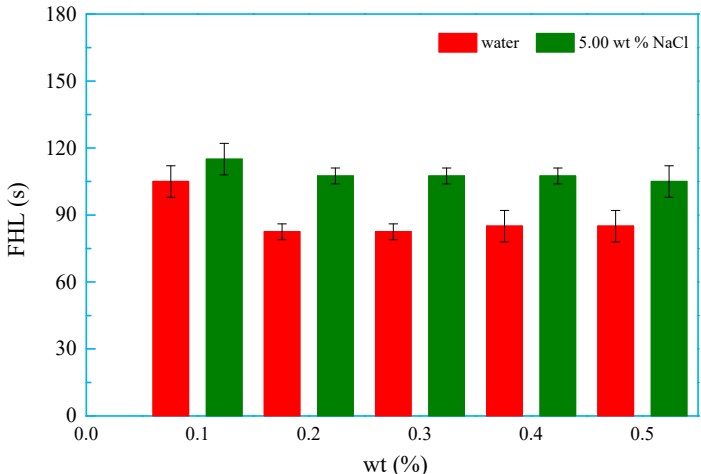

**Figure 15.** Foam half-lives of OSSF versus weight percentage in water and NaCl solution (T = 25 °C, free pH).

### 3.8.2. Defoaming Property

As seen in Figure 16, OSSF first accelerates the foam coalescence and collapse on the measuring cylinder wall but cannot reduce the defoaming time of the final thin layer on the surface. The defoaming process is clearly divided into the induction, acceleration, and attenuation periods. Figure 17 shows that the defoaming rate of OSSF first decreases sharply and then decreases to a constant when the weight percentage increases from 5.00 to 25.00 wt %. On the other hand, a decrease in defoaming time is observed with increasing weight percentage of OSSF. When the mass concentration of OSSF is greater than the critical micelle mass concentration, the formed micelles can adsorb and spread on the foam film and finally dissolve in the foam film. As the surface tension of OSSF is much lower than SDBS, the surface tension of OSSF adsorption points is less than the surrounding region, which can destroy the "self-healing" effect of the foam, and then can significantly accelerate merger and breakdown [29]. In the process of developing low permeability gas-condensate reservoirs, the defoaming effect of OSSF not only reduces the amount of defoamer, but also maintains the stability of drilling fluid column pressure, drilling fluid rheological properties, and pump efficiency.

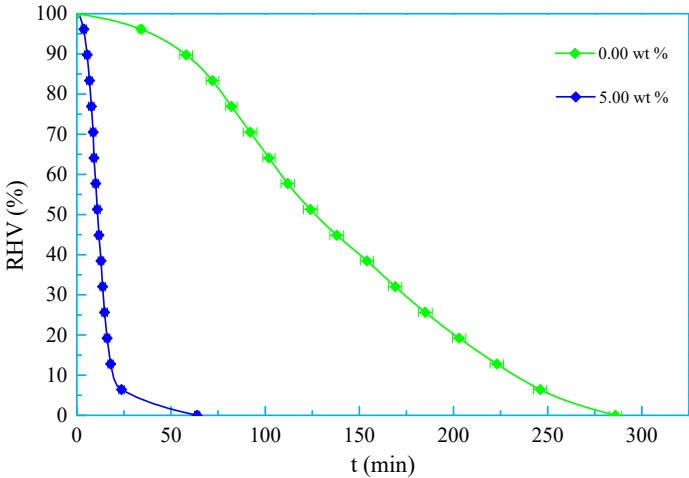

**Figure 16.** Relative foam volume of SDBS versus time at 5.00 wt % OSSF (T = 25 °C, free pH).

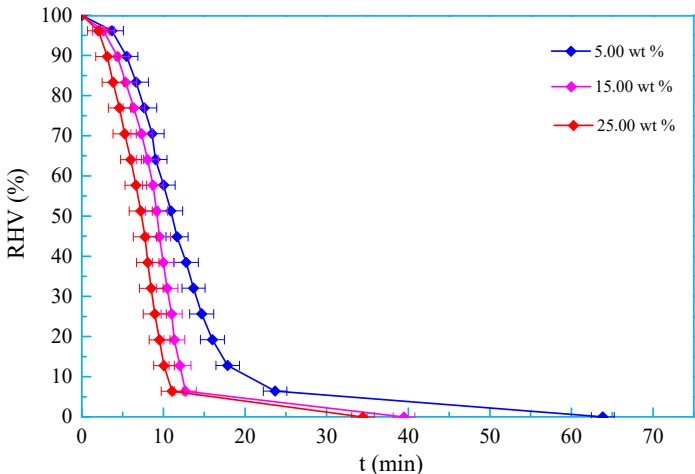

**Figure 17.** Relative foam volume of SDBS versus time at different weight percentages of OSSF (T = 25 °C, free pH).

### 3.8.3. Foam-Suppressing Property

Figure 18 shows that OSSF can decrease the foaming volume of SDBS and reduce SDBS foam stability in the induction, acceleration, and attenuation periods, which finally decreases the defoaming time of SDBS. Figure 19 shows that the foam-suppressing property of SDBS improves with increasing OSSF dosage from 0.1 to 0.5 wt %. OSSF and SDBS in water can simultaneously absorb on the gas–liquid interface. However, as the surface tension of the film adsorbed OSSF is much lower than the SDBS, the foam membrane thickness absorbed OSSF is thinner than SDBS, which ultimately accelerates the drainage of the foam membrane and gas diffusion through the foam membrane.

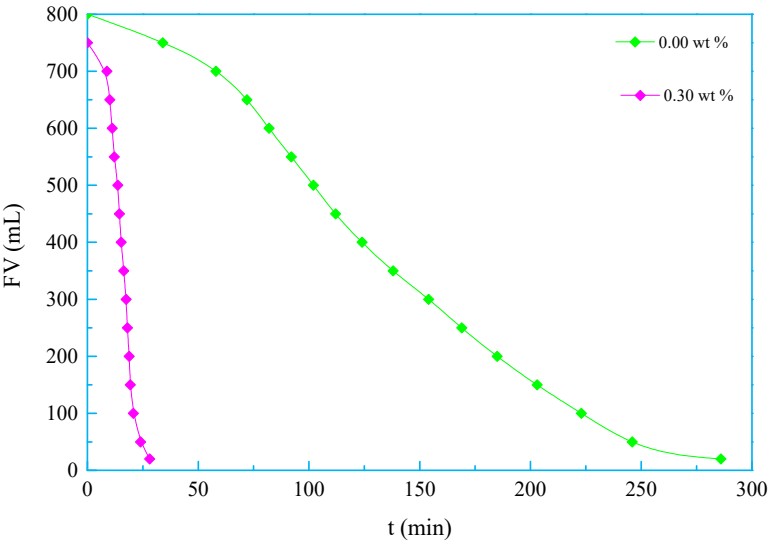

**Figure 18.** Foam volume of SDBS versus time at 0.30 wt % OSSF (T = 25 °C, free pH).

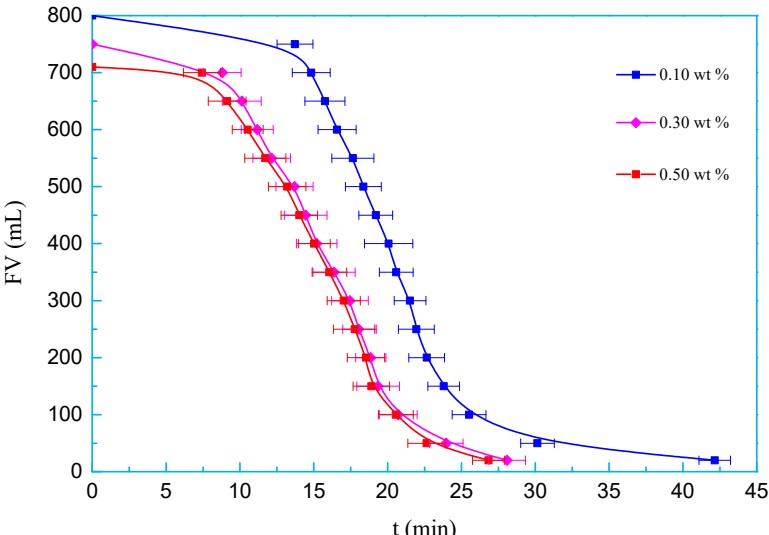

**Figure 19.** Foam volume of SDBS versus time at different weight percentages of OSSF (T = 25 °C, free pH).

## 4. Conclusions

The synthesis of the oligomeric silicone surfactant and the examination of the interfacial properties of low permeability gas-condensate reservoirs gave the following conclusions:

(1) A new kind of oligomeric silicone surfactant (OSSF) containing sulfonic acid groups was synthesized. The critical micelle mass concentration was 0.980 g/L and the critical surface tension was 20.631 mN/m. The surface tension of OSSF increased with increasing hot rolling temperature and decreased with the addition of NaCl, KCl, or $CaCl_2$.

(2) OSSF adsorption transferred the wettability of cores from water-wet to preferential gas-wet. A change in the OSSF adsorption layers' surface chemical composition occurred and exhibited lower interface energy than that of the cores. The adsorption isotherm of quartz sands changed from Langmuir type (L-type) to "double plateau" type (LS-type) in the OSSF solution due to the chemical adsorption at high temperature.

(3) The foaming volume and property of OSSF began to stabilize as the weight percentage increased. The presence of NaCl decreased the foaming volume and improved the OSSF foam stability. At the same time, OSSF decreased the initial foaming volume and stability in the induction period and accelerated sodium dodecyl benzene sulfonate (SDBS) formation.

**Author Contributions:** D.Y. and P.L. was the main supervisor of the research, J.Z. contributed in analysis of the results, X.Y. contributed in experiments design and conducted the experimental work and analyzed the results. R.W. wrote the manuscript, L.W. contributed in analysis of the results. S.W. contributed in analysis and discussion of the results and reviewing and editing of the manuscript.

**Funding:** This research was funded by thank Important and Special Project of China (No. 2016ZX05020-004) sponsored by the Ministry of Science and Technology of China.

**Conflicts of Interest:** The authors declare no conflict of interest.

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
