# Peer review of "Synthesis of Oligomeric Silicone Surfactant and its Interfacial Properties"

_applsci, doi:10.3390/app9030497_

Round 1
Reviewer 1 Report
This paper presents the interfacial activities of oligomeric surfactant with siloxane spacer and sulfonic group for improving production of gas from gas reservoirs. paper's title doesn't match with the content of the paper . In this work , we mainly observed evaluation of interfacial activities of OSSF and no data about condensate reservoirs or low permeable reservoirs are presented. I suggest to correct the title to be in line with real data.
Introduction needs to be more focus and express critically the other researchers work
The methodology is presented all over. It is better to provide work flow to explain what has been done.
There are plenty of typo , please make sure you do the proof reading before resubmitting this paper to journal
Results and discussion is divided into many subsections , 5 to 6 ss should be sufficient
Conclusion should be section 4 not 3 , and also conclusions not conclusion
Author Response
Dear editors and reviewers:
Thank you very much for reading our manuscript and reviewing it , which help to improve the quality of this manuscript and our scientific research capacity. We have checked the manuscript and revised it carefully according to the comments. We responded point by point to those comments as listed below and marked the revised portion in red in this paper. Which hoped these could make this paper more acceptable for publication.
We would like to express our great gratitude again to you and reviewers for the comments on our paper. I am looking forward to hearing from you.
Yours sincerely
Da Yin 
Corresponding author: 
Name: Jie Zhang 
mail: zhangjiedri@cnpc.com.cn
The following is a point-to-point response to the reviewer’ comments.
Comment 1: This paper presents the interfacial activities of oligomeric surfactant with siloxane spacer and sulfonic group for improving production of gas from gas reservoirs. Paper's title doesn't match with the content of the paper. In this work, we mainly observed evaluation of interfacial activities of OSSF and no data about condensate reservoirs or low permeable reservoirs are presented. I suggest to correct the title to be in line with real data.
Answer: Thank you very much for the positive feedback on our manuscript. we have corrected the title to the synthesis of oligomeric silicone surfactant and experimental study on effect of its interfacial properties.
Comment 2: Introduction needs to be more focus and express critically the other researchers work
Answer: we greatly appreciate your fair, encouraging and constructive comments to the literature review, I have made some modifications to this part in the paper. Our responses to the comment raised in the report are as follows:
Silicone surfactant is a kind of special surfactant, which is characterized by excellent thermal stability, high surface activity, outstanding ecological safety and biodegradability, which are mainly attributed to the feature of siloxane chain. Considerable research has been devoted to the synthesis and evaluation of anionic silicone surfactants. The sulfonate silicon surfactants are synthetized by free radical copolymerization of polysiloxanes and sulfonic acid monomers. Its hydrophobicity and oleophobicity increase with the chain length of polysiloxanes. The hydrophobic groups are crosslinked structure and have relatively low silicon content. As applying the free radical copolymerization, the molecular weight of synthetic surfactant is very high. Due to the hydrolysis of siloxane, the interfacial tension of the sulfonate trisiloxane surfactant prepared by hydrosilylation reaction increases after a long period of storage at normal temperature. The polyether trisiloxanes modified by the sulfonation reaction of the epoxy group exhibits excellent chemical stability such as acid-resistance, alkali-resistance and salt-resistance. But the siloxane chain is very short and they used the expensive and toxic chloroplatinic acid as catalysts for hydrosilation reaction. The phosphate fluorosilicone surfactant synthesized by hydrosilylation and esterification reaction of hydrogen-containing fluorine polysiloxane, polyethylene glycol monoallyl etherand phosphoric acid excellents low surface tension, critical micelle concentration and good resistance to acid, alkali and salt. As used the hydrosilicone oil as raw material, the molecular weight of synthetic surfactant is very high. they also used the expensive and toxic chloroplatinic acid as catalysts for hydrosilation reaction.
Comment 3: The methodology is presented all over. It is better to provide work flow to explain what has been done.
Answer: Thank you very much for the kind suggestions. We have provided detailed work flow of all test methods in characterization methods of our paper.
Comment 4: There are plenty of typo, please make sure you do the proof reading before resubmitting this paper to journal. Results and discussion is divided into many subsections, 5 to 6 ss should be sufficient, conclusion should be section 4 not 3, and also conclusions not conclusion.
Answer: we greatly appreciate you carefully read our paper and give us your positive and fair comments, I would try my best to correctly write this paper. Our responses to the detailed contact angle and surface energy are as follows:
Wetting is the ability of a liquid to maintain contact with a solid surface, which resulted from the intermolecular interactions when the liquid and solid are brought together. The energy of a solid relate to the reaction between atoms, molecules and ions of itself, including covalent bonds, ionic bonds, metallic bonds, van der Waals forces and hydrogen bonds. The cores mainly hold components together by chemical bonds. The surface energy of cores so high that the water can achieve complete wetting. However, the molecules OSSF adsorbed on the surface of cores mainly hold them together by physical forces, so the surface energy of cores reduces and they can be only partially wetted by water.
Although we knew the chemical composition of OSSF, the ratio of silicon, oxygen, carbon atoms in OSSF and in the rock, considering the porosity and nonuniformity of cores, we can’t estimate the adsorption amount of OSSF in the cores by solid-liquid interface and can only come to the conclusion that the OSSF can change surface component of cores.
Dear editors and reviewers:
Thank you very much for reading our manuscript and reviewing it , which help to improve the quality of this manuscript and our scientific research capacity. We have checked the manuscript and revised it carefully according to the comments. We responded point by point to those comments as listed below and marked the revised portion in red in this paper. Which hoped these could make this paper more acceptable for publication.
We would like to express our great gratitude again to you and reviewers for the comments on our paper. I am looking forward to hearing from you.
Yours sincerely
Da Yin 
Corresponding author: 
Name: Jie Zhang 
mail: zhangjiedri@cnpc.com.cn
The following is a point-to-point response to the reviewer’ comments.
Comment 1: This paper presents the interfacial activities of oligomeric surfactant with siloxane spacer and sulfonic group for improving production of gas from gas reservoirs. Paper's title doesn't match with the content of the paper. In this work, we mainly observed evaluation of interfacial activities of OSSF and no data about condensate reservoirs or low permeable reservoirs are presented. I suggest to correct the title to be in line with real data.
Answer: Thank you very much for the positive feedback on our manuscript. we have corrected the title to the synthesis of oligomeric silicone surfactant and experimental study on effect of its interfacial properties.
Comment 2: Introduction needs to be more focus and express critically the other researchers work
Answer: we greatly appreciate your fair, encouraging and constructive comments to the literature review, I have made some modifications to this part in the paper. Our responses to the comment raised in the report are as follows:
Silicone surfactant is a kind of special surfactant, which is characterized by excellent thermal stability, high surface activity, outstanding ecological safety and biodegradability, which are mainly attributed to the feature of siloxane chain. Considerable research has been devoted to the synthesis and evaluation of anionic silicone surfactants. The sulfonate silicon surfactants are synthetized by free radical copolymerization of polysiloxanes and sulfonic acid monomers. Its hydrophobicity and oleophobicity increase with the chain length of polysiloxanes. The hydrophobic groups are crosslinked structure and have relatively low silicon content. As applying the free radical copolymerization, the molecular weight of synthetic surfactant is very high. Due to the hydrolysis of siloxane, the interfacial tension of the sulfonate trisiloxane surfactant prepared by hydrosilylation reaction increases after a long period of storage at normal temperature. The polyether trisiloxanes modified by the sulfonation reaction of the epoxy group exhibits excellent chemical stability such as acid-resistance, alkali-resistance and salt-resistance. But the siloxane chain is very short and they used the expensive and toxic chloroplatinic acid as catalysts for hydrosilation reaction. The phosphate fluorosilicone surfactant synthesized by hydrosilylation and esterification reaction of hydrogen-containing fluorine polysiloxane, polyethylene glycol monoallyl etherand phosphoric acid excellents low surface tension, critical micelle concentration and good resistance to acid, alkali and salt. As used the hydrosilicone oil as raw material, the molecular weight of synthetic surfactant is very high. they also used the expensive and toxic chloroplatinic acid as catalysts for hydrosilation reaction.
Comment 3: The methodology is presented all over. It is better to provide work flow to explain what has been done.
Answer: Thank you very much for the kind suggestions. We have provided detailed work flow of all test methods in characterization methods of our paper.
Comment 4: There are plenty of typo, please make sure you do the proof reading before resubmitting this paper to journal. Results and discussion is divided into many subsections, 5 to 6 ss should be sufficient, conclusion should be section 4 not 3, and also conclusions not conclusion.
Answer: we greatly appreciate you carefully read our paper and give us your positive and fair comments, I would try my best to correctly write this paper. Our responses to the detailed contact angle and surface energy are as follows:
Wetting is the ability of a liquid to maintain contact with a solid surface, which resulted from the intermolecular interactions when the liquid and solid are brought together. The energy of a solid relate to the reaction between atoms, molecules and ions of itself, including covalent bonds, ionic bonds, metallic bonds, van der Waals forces and hydrogen bonds. The cores mainly hold components together by chemical bonds. The surface energy of cores so high that the water can achieve complete wetting. However, the molecules OSSF adsorbed on the surface of cores mainly hold them together by physical forces, so the surface energy of cores reduces and they can be only partially wetted by water.
Although we knew the chemical composition of OSSF, the ratio of silicon, oxygen, carbon atoms in OSSF and in the rock, considering the porosity and nonuniformity of cores, we can’t estimate the adsorption amount of OSSF in the cores by solid-liquid interface and can only come to the conclusion that the OSSF can change surface component of cores.
Reviewer 2 Report
In this m/s the authors have developed a new OSS to be used in low permeability gas condensate reservoirs. The abstract is well defining the project from the purpose to the results. Introduction part captures well the reason and motive behind this research paper. Synthesis steps for developing the OSS is well described and specific properties and relevant experiments in regards to the performance of the surfactant is well explained. Results are presented well and easy to understand and follow. Conclusion part is summarizing different part presented in the m/s and pointing out to the main factors affecting the process. This is not my area of expertise but as much as I understood, the m/s is well written and easy to follow and understand.
Author Response
Dear editors and reviewers:
Thank you very much for reading our manuscript and reviewing it , which help to improve the quality of this manuscript and our scientific research capacity. We have checked the manuscript and revised it carefully according to the comments. We responded point by point to those comments as listed below and marked the revised portion in red in this paper. Which hoped these could make this paper more acceptable for publication.
We would like to express our great gratitude again to you and reviewers for the comments on our paper. I am looking forward to hearing from you.
Yours sincerely
Da Yin 
Corresponding author: 
Name: Jie Zhang 
mail: zhangjiedri@cnpc.com.cn
Reviewer 3 Report
The main concern is regarding the method used in applying the solution to the rock surface in order to quantify its effect on the wetting characteristics. Soaking the rock in the solution and then evaporating the solvent results in an overestimation of the amount of surfactant that would adsorb onto the surface (compared to displacing the solution before drying). This was not acknowledged by the authors. In addition to overestimating the amount, it does not address the problem with desorption which renders the alteration of wettability to be a temporary rather than a permanent effect. My suggestion is that the authors address this issue by acknowledging this limitation for the applicability of those measurements. Possibly focusing on the flow-back of injected fluids would be more suitable than relating this to the enhanced production of condensate. The review of the literature does not address what surfactants are used in the oil and gas industry and what effect these have on interfacial tension.
The effect of temperature on interfacial tension is not expected. The increase in temperature usually results in a reduction in interfacial tension. This was not acknowledged or addressed by the authors given that they observed the opposite effect.
There were multiple statements that are not accurate in the paper. Those, along with some of the grammar and sentence structure issues are highlighted in the attached version of the paper.

Author Response
Dear editors and reviewers:
Thank you very much for reading our manuscript and reviewing it , which help to improve the quality of this manuscript and our scientific research capacity. We have checked the manuscript and revised it carefully according to the comments. We responded point by point to those comments as listed below and marked the revised portion in red in this paper. Which hoped these could make this paper more acceptable for publication.
We would like to express our great gratitude again to you and reviewers for the comments on our paper. I am looking forward to hearing from you.
Yours sincerely
Da Yin 
Corresponding author: 
Name: Jie Zhang 
mail: zhangjiedri@cnpc.com.cn
A point-to-point response to the reviewer’ comments are as follows:
Comment 1: The main concern is regarding the method used in applying the solution to the rock surface in order to quantify its effect on the wetting characteristics. Soaking the rock in the solution and then evaporating the solvent results in an overestimation of the amount of surfactant that would adsorb onto the surface (compared to displacing the solution before drying). This was not acknowledged by the authors. In addition to overestimating the amount, it does not address the problem with desorption which renders the alteration of wettability to be a temporary rather than a permanent effect. My suggestion is that the authors address this issue by acknowledging this limitation for the applicability of those measurements. Possibly focusing on the flow-back of injected fluids would be more suitable than relating this to the enhanced production of condensate.
Answer: Thank you very much for the kind suggestion. Our responses to the comments 1 are as follows:
This testing methodology of the contact angle and surface energy really has its defect, which could increase the amount of surfactant adsorbed onto the surface of cores. We acknowledged the limitations themselves in our paper.
We did some experiments about the change of contact angle with time, which can indicate that the alteration of wettability changes very little during a given period. The contact angles of water on the cores steeped into 0.50 wt% OSSF solution versus time are presented in Fig. 1 and Fig. 2. For the Yingxi cores and Haian cores, the contact angles of water on the cores adsorbed OSSF slowly decline to a stable value in one minute, which indicate that the OSSF absorbed on the cores can’t rapidly desorb in the water and the wettability alteration of cores could maintain in a certain period. We have added these content to our paper.
Yingxi
Haian
Fig. 1 Contact angles of water on the cores steeped into 0.50 wt% OSSF solution versus time
(room temperature, free pH)
Fig. 2 Contact angles of water on the cores steeped into 0.50 wt% OSSF solution versus time
(room temperature, free pH)
Studying the flow-back of injected fluids provides a good approach to the adsorbed amount of surfactant on the surface of cores. We will get into this aspect in more detail.
Comment 2: The review of the literature does not address what surfactants are used in the oil and gas industry and what effect these have on interfacial tension.
Answer: Thank you for this nice suggestion. The review of the application and the effect of surfactants in the oil and gas industry have summarized in the introduction. Our specific responses to this comment 2 are as follows:
Surfactants are organic compounds composed of hydrophilic polar groups and hydrophobic non-polar groups. A small amount of them can significantly change its interfacial properties. At present, the commonly used surfactants in oil and gas development include hydrocarbon surfactant and fluorocarbon surfactants. Hydrocarbon surfactants can reduce surface tension between gas and liquid to about 35~30 mN/m. As generally cheap and low pollution, they are widely used in emulsion, foaming and reducing the capillary pressure such as OP-10, ABS and ABSN. However, they can’t change the wettability of reservoirs from water-wet to preferential gas-wet by adsorption. Fluorocarbon surfactants has good surfactivity, which can decrease the surface tension between gas and liquid to below 20 mN/m. They also have unique effects on defoaming and wettability alteration, but they are expensive and not environment-friendly, which limit the extensive applications.
Comment 3: The effect of temperature on interfacial tension is not expected. The increase in temperature usually results in a reduction in interfacial tension. This was not acknowledged or addressed by the authors given that they observed the opposite effect.
Answer: Thank you very much for this positive feedback on our manuscript. Our responses to the comment 3 are as follows:
Hot rolling test is generally adopted to study the influences of high temperature and shear on the stability of drilling fluid component, Figure 2 is the surface tension of OSSF versus mass concentration under various hot rolling temperature. Those OSSF solution rolled at different temperature in the roller oven but their surface tension was all measured at 25 ℃, so these trends can’t explain the effect of temperature on interfacial tension of OSSF but the effect of temperature on the stability of interfacial performance of OSSF. We found that the surface tension of OSSF solution at 25 ℃ only increase slightly after hot rolling, which can show the good stability of interfacial performance of OSSF. However, they decrease with the increase of hot rolling temperature.
Comment 4: There were multiple statements that are not accurate in the paper. Those, along with some of the grammar and sentence structure issues are highlighted in the attached version of the paper.
Answer: we greatly appreciate your positive and careful comments, I would try my best to make a clear and concise statement.

Round 2
Reviewer 1 Report
Accepted after final proofreading
Author Response
The article has been carefully proofread.
Reviewer 3 Report
The responses and corrections on the technical comments are sufficient. The paper still has grammatical mistakes, including ones that were highlighted in the version I attached in the last revision.
Author Response
We have checked the manuscript and revised grammatical mistakes carefully.